# Insignificant effect of climate change on winter haze pollution in Beijing

Lu Shen[1], Daniel J. Jacob[1], Loretta J. Mickley[1], Yuxuan Wang[2,3], Qiang Zhang[3]

[1]School of Engineering and Applied Sciences, Harvard University, Cambridge, MA 02138, USA
[2]Department of Earth and Atmospheric Sciences, University of Houston, Houston, TX, USA
[3]Department of Earth System Science, Tsinghua University, Beijing, China

*Correspondence to*: Lu Shen (lshen@fas.harvard.edu)

**Abstract.** Several recent studies have suggested that 21st-century climate change will significantly worsen the meteorological conditions leading to very high concentrations of fine particulate matter ($PM_{2.5}$) in Beijing in winter (Beijing haze). We find that 81% of the variance in observed monthly $PM_{2.5}$ during 2010-2017 winters can be explained by a single
meteorological mode, the first principal component (PC1) of the 850 hPa meridional wind velocity (V850) and relative humidity (RH). V850 and RH drive stagnation and chemical production of $PM_{2.5}$, respectively, and thus have a clear causal link to Beijing haze. PC1 explains more of the variance in $PM_{2.5}$ than either V850 or RH alone. Using additional meteorological variables does not explain more of the variance in $PM_{2.5}$. Therefore PC1 can serve as a proxy for Beijing haze in the interpretation of long-term climate records and in future climate projections. Previous studies suggested that shrinking
Arctic sea ice would worsen winter haze conditions in eastern China, but we show with the PC1 proxy that Beijing haze is correlated with a dipole structure in the Arctic sea ice rather than with the total amount of sea ice. Beijing haze is also correlated with dipole patterns in Pacific sea surface temperatures (SSTs).   We find that these dipole patterns of Arctic sea ice and Pacific SSTs shift and change sign on interdecadal scales, so that they cannot be used reliably as future predictors for the haze. Future 21st-century trends of the PC1 haze proxy computed from the CMIP5 ensemble of climate models are
statistically insignificant. We conclude that climate change is unlikely to significantly offset current efforts to decrease Beijing haze through emission controls.

## 1 Introduction

Severe wintertime haze pollution in Beijing caused by particulate matter finer than 2.5 μm aerodynamic diameter ($PM_{2.5}$) is a major threat to public health. Wintertime (December- January-February, DJF) $PM_{2.5}$ in Beijing averaged 110 μg m$^{-3}$ during
2009-2015 (Cai et al., 2017), more than four times the recommended 24-hour standard (25 μg m$^{-3}$) of the World Health Organization (WHO 2006). This Beijing haze is caused by a combination of high emissions and poor ventilation. The most severe haze events occur during sustained stagnant conditions with high humidity (Wang Y.S. et al., 2014).

Several recent studies (Wang et al., 2015; Cai et al., 2017; Zou et al., 2017; Li et al., 2018) have suggested that climate change will worsen Beijing haze, offsetting the major efforts currently underway to reduce emissions. Wang et al. (2015) found that shrinking Arctic sea ice could lead to a northward shift of cyclone tracks and weaker Rossby wave activity in eastern China, resulting in a more stable atmosphere. Zou et al. (2017) found that continued Arctic sea ice loss together with more extensive snowfall in Eurasia could worsen ventilation in the East China Plain. The synoptic-scale variable most directly affecting Beijing haze is the strength of the Eastern Asia Winter Monsoon (EAWM), which ventilates Beijing with clean air from the north. By analyzing $PM_{2.5}$ observations in Beijing during the winters of 2010-2017, Cai et al. (2017) found that haze events are associated with weakened northerly wind at 850 hPa, weakened northwesterly wind at 500 hPa, and enhanced temperature difference between 250 hPa and 850 hPa. Using climate projections for these three variables from the IPCC Coupled Model Intercomparison Project Phase 5 (CMIP5) (Taylor et al., 2012) model ensemble, they inferred that the days conducive to severe haze events ($> 150$ μg m$^{-3}$) would increase by 50% by 2100 due to climate change under the business-as-usual Representative Concentration Pathway (RCP) 8.5 climate change scenario (Cai et al., 2017). In further simulations with a climate model, Li et al. (2018) specifically related these changes in the EAWM to increasing greenhouse gases.

Here we re-examine the effect of climate change on Beijing winter haze by bringing in several meteorological factors not adequately considered in previous work. First is the effect of relative humidity (RH), which is not only an indicator of stagnation but also drives $PM_{2.5}$ production through aqueous-phase heterogeneous chemistry (Wang Y.S. et al., 2014; Wang Y.X. et al., 2014; Woo and McNeill, 2015; Shen et al., 2017; Leung et al., 2018; Zhang et al., 2018). Second is the complexity of the relationship between Beijing haze and Arctic sea ice, which we will show reflects not the total amount of ice but its spatial distribution. Third is the role of sea surface temperatures (SSTs) in the Pacific and Indian Ocean, known to affect wintertime weather in the North China Plain (Annamalai et al., 2005; Kim et al., 2016). Fourth, as we will show, is the multi-decadal variability in the relationships of Arctic sea ice and SSTs to the regional conditions conducive to Beijing haze. We conclude that the effect of climate change on Beijing haze is far more uncertain than reported in previous studies and most likely inconsequential for haze control strategies.

**2 Data**

We base our analysis on the 2010-2017 continuous record of $PM_{2.5}$ available from the U.S. Embassy in Beijing. The measurements are done with a beta attenuation monitor at fixed RH (San Martini et al., 2015) (http://www.stateair.net/web/historical/1/1.html). Li et al. (2018) have shown that this dataset can be considered representative of mean $PM_{2.5}$ concentrations in Beijing by comparing with observations from other 12 sites. For the in-situ meteorological observations, we use RH and surface wind speed from the NOAA Global Summary of the Day (GSOD)

network. For the vertical temperature gradient, we rely on the Integrated Global Radioradiosonde Archive (IGRA) (Durre et al., 2008). For the east-west and north-south wind velocities at 850 and 500 hPa, we use the National Centers for Environmental Prediction (NCEP) Reanalysis 1, mapped onto the 2.5° grid resolution (Kalnay et al., 2006). Velocities are positive northward and eastward. For SSTs, we rely on the NOAA Extended Reconstructed SSTs (ERSST v4) (Huang 2016).

For the sea ice concentrations (SICs), we use Hadley Centre Sea Ice and Sea Surface Temperature data set (HadISST) (Rayner et al., 2003). The SIC is unitless, ranging from 0 to 1. The sea ice area is defined as the area of each gridcell multiplied by its SIC.

To project the effects of climate change on $PM_{2.5}$ air quality, we use meteorological fields from climate models participating in the Coupled Model Intercomparison Project Phase 5 (CMIP5) and following the RCP8.5 scenario (Taylor et al., 2012).

We use an ensemble of 32 models that provide monthly values and 21 models that also provide daily values. The CMIP5 data are archived at a horizontal resolution of ~200 km, and the details of these models can be found in Table S2.

## 3 Relationship of Beijing haze to local/regional weather variables

Figure 1a shows the correlations of DJF monthly mean $PM_{2.5}$ in Beijing with the meridional wind velocity at 850 hPa from the NCEP reanalysis (Kalnay et al., 1996). There is a strong relationship as previously pointed out by Cai et al (2017). The

15 wind is consistently from the north, and the corresponding velocity is negative (southward), therefore the correlation with $PM_{2.5}$ is positive (high $PM_{2.5}$ is associated with weak or northerly wind). Maximum correlation is for the rectangular domain to the north of Beijing, and we will refer to V850 in what follows as the mean meridional wind velocity at 850 hPa within that domain. Figure 1b shows in addition a strong correlation of monthly $PM_{2.5}$ with local relative humidity measured at the Beijing airport (RH in what follows). We use the Beijing airport measurement because the NCEP analysis is not reliable for

relative humidity (Fig. S1). The Beijing airport measurements of RH are regionally representative a shown by strong correlation ($R = 0.92$) with measurements from other sites in the North China Plain (Fig. 2). We find that other meteorological variables previously linked to winter haze, including the difference between the 500 hPa zonal winds north and south of Beijing (95°E-130°E, 47.5°N-55°N vs. 120°E-145°E, 27.5°N-35°N) ($\delta$U500, Fig. S2), and vertical potential temperature gradients between 925 and 1000 hPa ($\delta T_{925-1000}$) (Zou et al., 2017), have lower correlations with monthly mean

$PM_{2.5}$ than V850 and RH. (Fig. S3).

Figure 1c shows the correlation of Beijing $PM_{2.5}$ with the first principal component (PC1) of monthly mean V850 and RH. PC1 is the sum of V850 and RH after normalization of units:

$$PC1 = \frac{1}{\sqrt{2}}(\frac{V850 - V850_{mean}}{V850_{std}} + \frac{RH - RH_{mean}}{RH_{std}}) \tag{1}$$

where V850$_{mean}$ and RH$_{mean}$ refers to the temporal mean, and V850$_{std}$ and RH$_{std}$ refers to the standard deviation. We find a much stronger correlation of PM$_{2.5}$ with PC1 ($R = 0.90$) than with either V850 ($R = 0.81$) or RH ($R = 0.78$). V850 and RH are moderately correlated ($R = 0.60$). PC1 includes information from both variables in a way that can better characterize the main weather mode related to the haze pollution. Figure 1d shows the 2010-2017 normalized time series of DJF monthly

mean PM$_{2.5,}$ V850, RH, and PC1, illustrating how the relationship of PM$_{2.5}$ with PC1 is much stronger than with either V850 or RH. We also conducted principal component analyses using additional meteorological variables including surface wind speed, $\delta$U500, and $\delta$T$_{925-1000hPa}$, and find that the first principal components do not resolve additional PM$_{2.5}$ variability (Table S1). The strong correlation of PC1 with PM$_{2.5}$ has a clear physical basis because weak winds correspond to stagnation events and high RH promotes chemical PM$_{2.5}$ formation. In what follows we will use the PC1 of V850 and RH as meteorological

proxy for PM2.5 in order to analyse longer historical records than the 2010-2017 period of PM$_{2.5}$ observations and in order to make projections of PM$_{2.5}$ response to future climate change.

Our PC1 predictor of PM$_{2.5}$ shows large interannual variability over 1973-2017 but no significant trend (Fig. 1c), and neither do the principal components constructed from additional meteorological variables (Table S1, Fig. S4). Zou et al. (2017) previously defined a pollution potential index for the East China Plain (30°N-41°N, 112°E-122°E) on the basis of surface

wind speed and the temperature gradient between 925 and 1000 hPa, both taken from the NCEP reanalysis (Kalnay et al., 1996). They found a significant increase of this index during 1980-2017, driven by decreasing wind speed and increasing temperature gradient in the lower troposphere.  However, the NCEP reanalysis trends for these two variables appear to be spurious. When using in-situ observations from the 13 North China Plain sites, we find that the surface wind speed decreases from 1980 to early 2000s but then increases after 2005 (Fig. S5), in phase with the EAWM strength (Ding et al., 2014) that

also increases after 2005. Overall the trend in surface wind speed during 1980-2017 is insignificant. The meteorological radiosonde data in the North China Plain show a significant decreasing temperature gradient between 925 and 1000 hPa from 1993 to 2017 (Fig. S6a), in contrast to the NCEP data (Fig. S6b).

## 4 Relationship of Beijing haze to Arctic sea ice and sea surface temperatures

Previous studies have suggested that Arctic sea ice melting would worsen haze pollution in eastern China (Wang et al., 2015; Zou et al., 2017). But we find the correlation of the PC1 Beijing haze proxy with total Arctic sea ice cover (SIC) to be insignificant ($R = -0.08$). This is consistent with Yin and Wang et al (2017), who also found insignificant relationship of North China Plain haze pollution in December with total Arctic sea ice. Instead, we find that the relationship of PC1 with SIC is more complicated and displays a dipole pattern, with locations shifting in different months (Fig. 3a-b, Fig. S7). Figure

3a shows the slopes of Arctic sea ice in the preceding autumn regressed onto PC1 in winter, displaying positive slopes in the Laptev Sea (90°E-150°E) but negative slopes elsewhere. Here we define the SON-$\Delta$SIC as the difference of total sea ice area

between regions with significant positive and negative slopes. The correlation coefficient of PC1 with SON-$\Delta$SIC is 0.60. Previous research has identified a dipole Arctic sea ice pattern in the SON and shown that this pattern can excite a stationary Rossby wave train propagating southeastward to Eurasia (Chen et al., 2014) with predictive capability for the EAWM (Chen et al, 2014; Sun et al., 2016) and for winter haze frequency in the North China Plain (Yin and Wang, 2016a).

Haze pollution in Beijing is also modulated by SST variability in the Pacific (Yin and Wang, 2016b) and Indian Ocean (Fig. 3c-d), implying that the projection of future haze at least needs to consider changes of both SSTs and SICs. Figure 3c shows the slopes of SSTs in the preceding autumn (SON) regressed onto the PC1 haze proxy in winter during 1973-2017. We see a dipole pattern over the North Pacific with cool SST anomalies for the western Pacific and warm anomalies for the eastern Pacific. From autumn to winter the correlations shift to the tropics with a dipole pattern of warm SST anomalies for the

Indian Ocean and cool anomalies for the western tropical Pacific (Fig. 3d, Fig. S8). Yin and Wang (2016b) found that cool SSTs in the subtropical western Pacific in the previous autumn could excite a Rossby wave train that weakens the prevailing northerly wind over the North China Plain. Li et al. (2015) showed that anomalous southerly wind along the coast of eastern China could also be induced by a warm Indian Ocean, transporting moisture to eastern China. We also identify enhanced relative humidity across much of the eastern China associated with warmer SSTs in the Indian Ocean (Fig. S9). To

characterize the SST dipole pattern in the North Pacific in Fig. 3c, we define SON-$\Delta$SST as the average SST difference between western Pacific and Northeast Pacific. Using these two metrics, SON-$\Delta$SST and SON-$\Delta$SIC, we can predict about ~50% of the variance of PC1 one season in advance using a linear model. Using a 7-year moving average to detrend the timeseries yields a correlation coefficient of 0.69 between reconstructed and predicted PC1 during 1973-2017, and a correlation coefficient of 0.76 between observed PM$_{2.5}$ and predicted PC1 based on a limited history during 2010-2017 (Fig.

3e).

    However, we find that these relationships of PC1 with SST and SIC shift on timescales of decades, compromising their use to project future PM$_{2.5}$ changes. Both V850 and RH over 1950-2017 can be obtained from the NCEP reanalysis, while local RH measurements in Beijing date back only to the mid-1970s. However, we have little confidence with the NCEP RH because it is only weakly correlated with local measurements (Fig. S1), and so we focus on V850 going back to 1950. Figure

4a-b shows the slopes of SICs in the preceding SON regressed onto DJF V850 during 1950-1979 and 1980-2017. We choose the year 1980 as the boundary between these two timeframes because the SIC observations before 1980 are not validated with satellite observations (Rayner et al., 2003). Previous studies have also linked the EAWM with variations of the Arctic Oscillation (AO). As with Arctic sea ice, however, the EAWM-AO relationship displays large inter-decadal variability, with a strengthening since the 1980s (Li et al., 2014). The decadal oscillation of the EAWM-AO relationship could be attributable

to the location of the East Asian jet stream (Li et al., 2014). Similarly, the relationship of DJF V850 and SSTs also displays very different dipole patterns during 1950-1976 and 1977-2017 (Fig. 4c-d), including a weaker relationship with El Niño–

Southern Oscillation (ENSO) in the more recent record. This result is consistent with previous studies that identified a weakened EAWM-ENSO relationship after the mid-1970s (Kim et al., 2016; Wang and He, 2012, He and Wang, 2013). Several mechanisms have been proposed to explain this including a phase shift of the Pacific Decadal Oscillation (PDO) (Wang et al., 2008), the combined effects of Atlantic Multidecadal Oscillation and PDO in modulating the North Pacific oscillation (He and Wang, 2013), and the varying location and intensity of the northwest Pacific anticyclone combined with a Walker circulation between the eastern and central Pacific (Kim et al., 2016). Using the 20[th] Century Reanalysis, He and Wang (2013) found that this low-frequency oscillation in the EAWM-ENSO relationship has a period of 50 years, and that it may have recovered in the past decade as the PDO has shifted to a negative phase and the Walker circulation has strengthened.

Thus, we conclude that future climate projections of SIC or SSTs do not allow confident prediction of the meteorological conditions conducive to Beijing haze, because the relationships of the haze with these variables are complicated and fluctuate on multidecadal scales, apparently in response to low-frequency climate modes. The relationship with SIC depends on a dipole structure in the sea ice, rather than the total amount of sea ice, so that there is no simple link between shrinking Arctic sea ice and increased haze pollution in Beijing.

**5 Insignificant effect of 21$^{st}$-century climate change on Beijing haze**

The strong physical basis relating our PC1 meteorological proxy to Beijing haze suggests that it can be used to project the effect of future climate change. Figure 5a-b shows mean 2080-2099 vs. 2000-2019 trends in 850 hPa meridional wind velocity (V850) and RH from the ensemble of 32 CMIP5 models following the RCP8.5 business-as-usual scenario. The 850 hPa meridional wind shows no consistent trend over China (Fig. 5a), as also reported by Xu et al. (2016). We find a decrease of RH by 1-6% across much of East Asia (Fig. 5b), likely due to (1) the increasing land-ocean thermal contrast (Byrne and O'Gorman, 2013) and (2) enhanced subsidence at mid-latitudes as a consequence of the expanding Hadley circulation (Lau and Kim, 2015). The RH changes in Beijing do not reach statistical significance, however. Using projected changes in PC1 from the individual CMIP5 models, together with the PM$_{2.5}$ vs. PC1 relationship of Fig. 1c, we find that climate change alone would decrease the DJF mean PM$_{2.5}$ in Beijing by -0.68 ± 11.7 μg m$^{-3}$ from 2000-2019 to 2080-2099, where the mean and standard deviation apply to the ensemble of CMIP5 models (Fig. 5c). The insignificant mean and the large spread imply little confidence in the projections from individual models including in the sign.

We pointed out previously how alternative principal component predictors of Beijing haze could be derived using additional meteorological variables (i.e., $\delta T_{925-1000}$, surface wind speed). We find that these alternative principal components similarly show no significant trend from 2000-2019 to 2080-2099. The associated mean changes in PM$_{2.5}$ from the ensemble of

CMIP5 models range from -1.40 to 1.03 μg m$^{-3}$ for the different principal components (Fig. 6, Fig. S10), or about ±1% of mean 2010-2017 concentrations.

Our results apply to monthly mean PM$_{2.5}$ concentrations, but Pendergrass et al. (2018) used the same meteorological variables with an extreme value point process model to project the effect of climate change on the occurrence of severe

wintertime PM$_{2.5}$ pollution events (24-h average > 300 μg m$^{-3}$) in Beijing. They similarly find low V850 and high RH to be the two best predictor variables for these extreme haze events. They show no significant effect of climate change in the frequency of haze events for the RCP8.5 scenario, using the same ensemble of CMIP5 climate models as here, and a decrease for the alternative RCP4.5 scenario due to decreasing RH under low wind speed conditions. They conclude that climate change would most likely decrease the frequency of severe haze events in Beijing.

One remaining question is why our results are not consistent with Cai et al. (2017), who previously found a 50% increase by 2100 in the number of days conducive to severe Beijing haze events (> 150 μg m$^{-3}$), using 15 CMIP5 models applied to the RCP8.5 climate forcing scenario. They based their conclusions on projected increases in V850, increased temperature differences between 850 hPa and 250 hPa ($\delta T_{250-850}$), and increases in the meridional gradient of the 500 hPa zonal wind ($\delta U500$). We confirm these trends in $\delta T_{250-850}$ and $\delta U500$ in the more complete ensemble of 32 CMIP5 models, but do not

find an increase in V850. Cai et al. (2017) showed that higher $\delta U500$, characterized by weakened zonal flow south of Beijing but strengthened zonal flow north of Beijing, could lead to higher V850. We confirm this relationship both in the NCEP reanalysis during 2000-2019 (Fig. S2) and in the CMIP5 models for the 21$^{st}$ century (Fig. S11a), but the trend in $\delta U500$ does not appear to drive a trend in V850 (Fig. S11b-c). Similarly, the trend in $\delta T_{250-850}$ does not appear to drive a trend in $\delta T_{925-1000}$ (Fig. S10a). Most importantly, Cai et al. (2017) did not consider RH as a predictor variable for the

occurrence of extreme haze even though it plays a critical role and is only moderately correlated with V850. Indeed, Pendergrass et al. (2018) showed that they could reproduce the increase in the frequency of extreme haze events found by Cai et al. (2017) by omitting RH as a predictor variable in their point process model, but that the resulting model did not properly fit the observed extreme value data.

**6 Conclusions**

Several recent studies have suggested that 21$^{st}$-century climate change will greatly worsen the meteorological conditions leading to severe wintertime PM$_{2.5}$ pollution in Beijing (winter haze). We find here that this not the case.

Meridional wind velocity (V850) and relative humidity (RH) are strong predictors of the variability of monthly mean PM$_{2.5}$ in wintertime Beijing (Cai et al., 2017; Wang Y.S. et al., 2014; Wang Y.X. et al., 2014, Leung et al., 2018). We find that ~80% of the PM$_{2.5}$ variance can be explained by the first principal component (PC1) of these two variables, i.e., the sum of their

normalized values. There is a strong physical basis for this relationship, as high V850 (weak northerly or southerly wind) is associated with stagnation, and high RH promotes chemical production of $PM_{2.5}$. V850 and RH are only moderately correlated (R = 0.60). Thus PC1 is the better proxy for the meteorological conditions leading to Beijing haze. Principal components constructed from additional meteorological variables used in previous studies, including the 500 hPa zonal wind dipole and vertical temperature gradient, do not improve predictability and show similar results.

Previous studies suggested that Arctic sea ice melting would worsen haze pollution in eastern China (Wang et al., 2015; Zou et al., 2017), but we find no correlation between total Arctic sea ice and the PC1 haze proxy. PC1 is instead significantly correlated to a dipole pattern in Arctic sea ice in the preceding autumn that is more difficult to interpret in terms of climate change. We find that PC1 is also associated with cool SSTs in the northwestern Pacific, consistent with a previous study (Yin and Wang, 2016b). However, we also find that these associations of PC1 with sea ice and SSTs shift and change sign on multidecadal time scales, possibly in concert with other climate oscillations. Thus it is not possible to use these large-scale climate variables in future projections of the effect of climate change on Beijing haze.

We went on to examine the 2000-2100 regional projections of V850, RH, and PC1 in the ensemble of IPCC CMIP5 models as a more direct inference of the effect of climate change on Beijing haze. We find that changes in $PM_{2.5}$ inferred from changes in PC1 are inconsistent in sign across models and overall very small (~1%). It appears from our results that the effect of climate change on Beijing winter pollution is insignificant.

**Acknowledgments.** We thank Yufei Zou (Georgia Tech University) and Jintai Lin (Peking University) for fruitful discussion on the mechanism of large-scale climate patterns related to Beijing haze. This work was funded by the Harvard Global Institute (HGI).

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

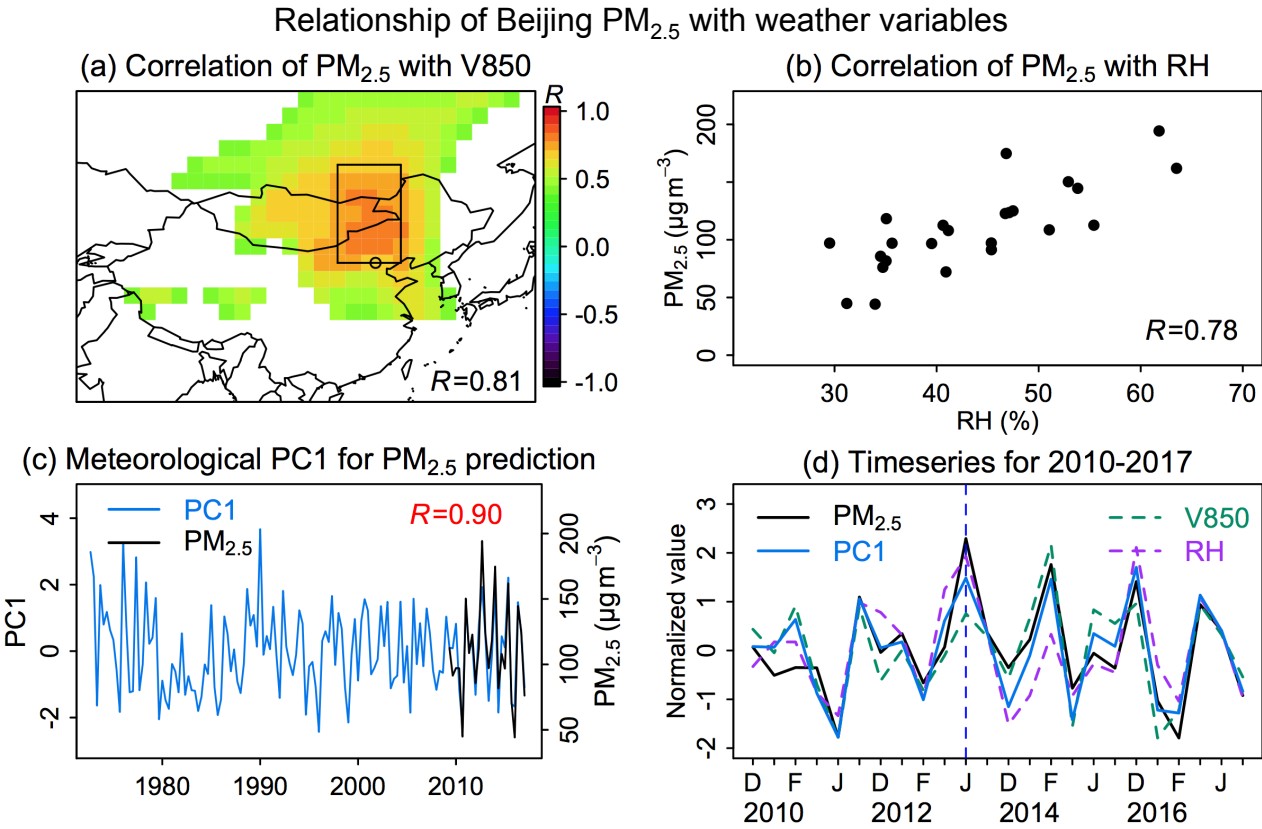

**Figure 1**. Relationships of wintertime monthly mean $PM_{2.5}$ concentrations in Beijing (DJF 2010-2017 record) with local/regional meteorological variables. (a) Correlations with the 850 hPa meridional wind velocity in Beijing and the surrounding region. The open circle denotes Beijing. The maximum correlation is for the rectangular box, which we use to define V850 in the text. The correlation coefficient $R$ of monthly $PM_{2.5}$ and V850 in that box is shown inset. (b) Relationship with local relative humidity (RH) measured at Beijing airport, with the correlation coefficient shown inset. (c) Timeseries of the first principal component (PC1) of V850 and RH for the 1973-2017 preiod (see Equation (1) for definition of PC1). The time series of observed $PM_{2.5}$ during 2010-2017 is also shown. The correlation coefficient of PC1 and $PM_{2.5}$ during 2010-2017 is shown inset. (d) Monthly mean timeseries for 2010-2017 of PC1 and normalized $PM_{2.5}$, V850, and RH, where the normalization is relative to the 2010-2017 means.

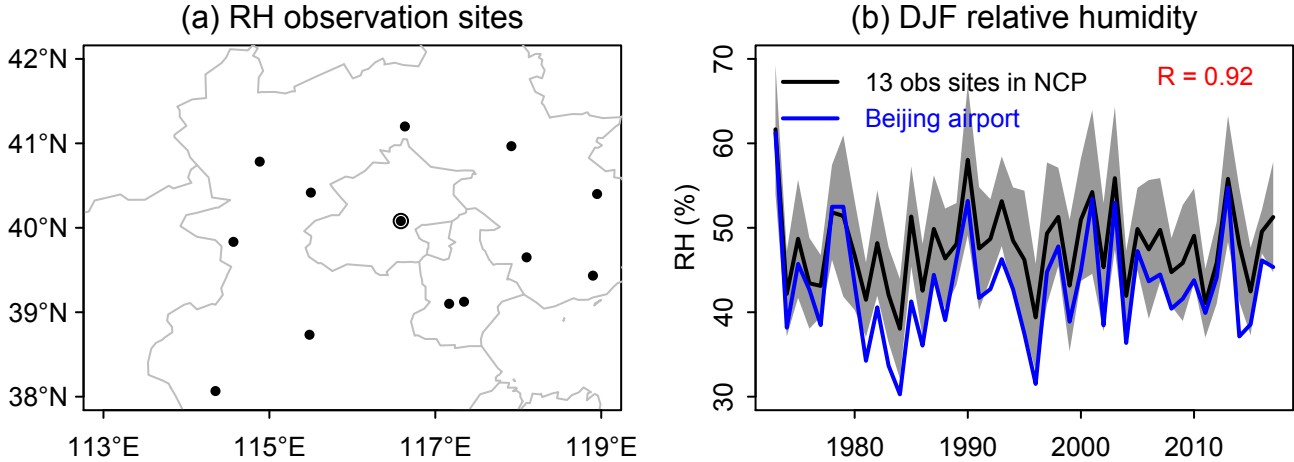

**Figure 2**. Representativeness of relative humidity (RH) observed at Beijing airport and used in this work for correlation to PM$_{2.5}$. (a) Location of Beijing airport and the 12 surrounding sites of the NOAA Global Summary Of the Day (GSOD) network in the North China Plain (NCP). (b) Timeseries of DJF 1973-2017 mean relative humidity (RH) at the Beijing airport site and for the ensemble of the 13 NCP sites. Shaded area denotes one standard deviation. The correlation coefficient (R) of RH at Beijing airport with the North China Plain (NCP) average is 0.92.

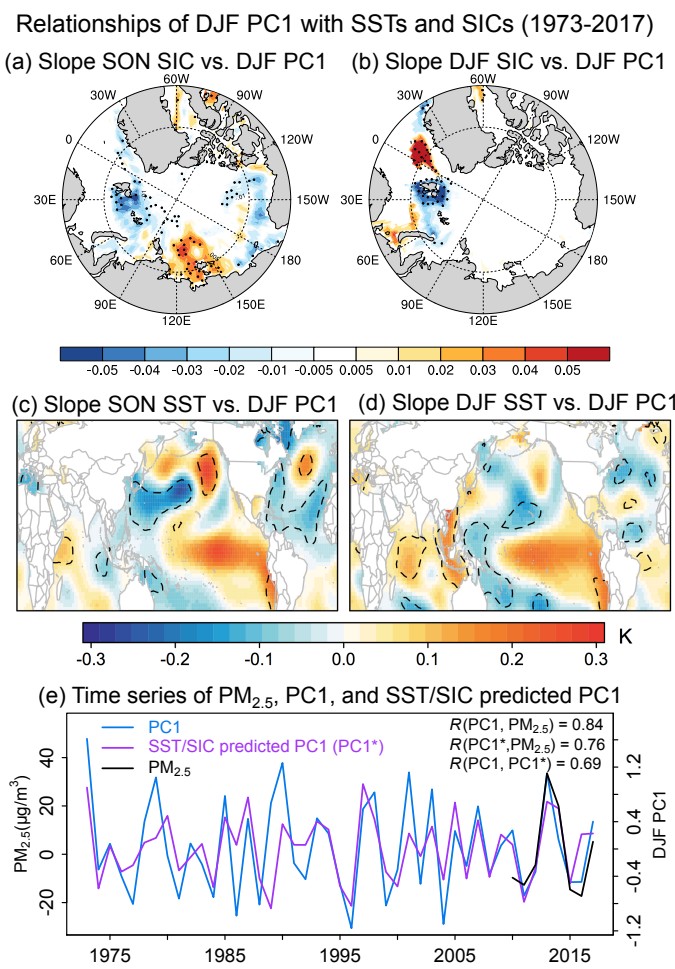

**Figure 3**. Relationships of sea ice concentrations (SICs) and sea surface temperatures (SSTs) with Beijing winter haze diagnosed from the first meteorological principal component PC1 (Equation (1)) for 1973-2017. The top panels show the slopes of the linear regressions for SICs in the (a) preceding autumn and (b) current winter vs. normalized PC1. Gridboxes with statistically significant correlations ($p < 0.1$) are stippled. (c-d) The middle panels show the corresponding slopes of the linear regressions for SSTs. The dashed contour lines enclose regions in which correlations are statistically significant ($p < 0.1$). (e) The bottom panel shows the time series of observed wintertime mean $PM_{2.5}$ concentrations in Beijing, PC1, and the predicted PC1 (PC1*) from SST and SIC (see text). The correlation coefficients of different timeseries are shown inset. For all panels, data are detrended by subtracting the 7-year moving average. The correlation coefficient of PC1 with $PM_{2.5}$ ($R = 0.84$) is lower than in Fig. 1 ($R = 0.90$) because it is calculated here from seasonal rather than monthly average data.

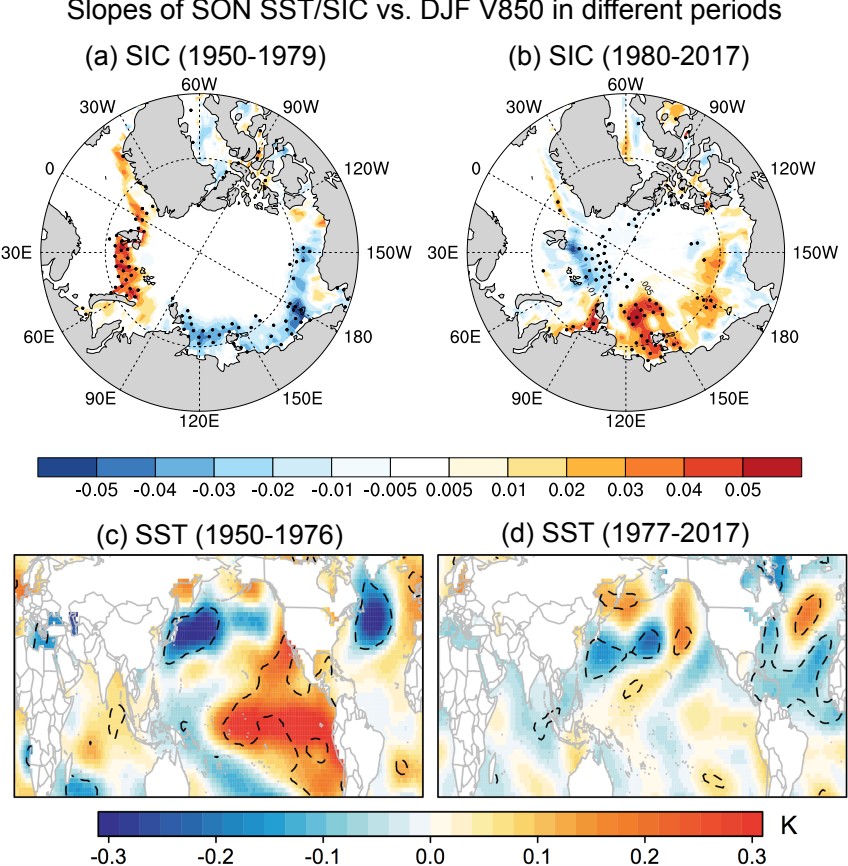

**Figure 4**. Multidecadal shifts in the relationships between the meridional wind over northern China in winter (V850) with sea ice concentrations (SICs) and sea surface temperatures (SSTs) in the preceding autumn (SON). The top panels show the regression slopes of SON SICs vs. normalized DJF V850 during (a) 1950-1979 and (b) 1980-2017. Gridboxes with statistically significant correlations (p<0.1) are stippled. (c-d) The bottom panels show the same for SSTs. The dashed contour lines enclose regions in which correlations reach statistical significance (p < 0.1). For all panels, data are detrended by subtracting the 7-year moving average.

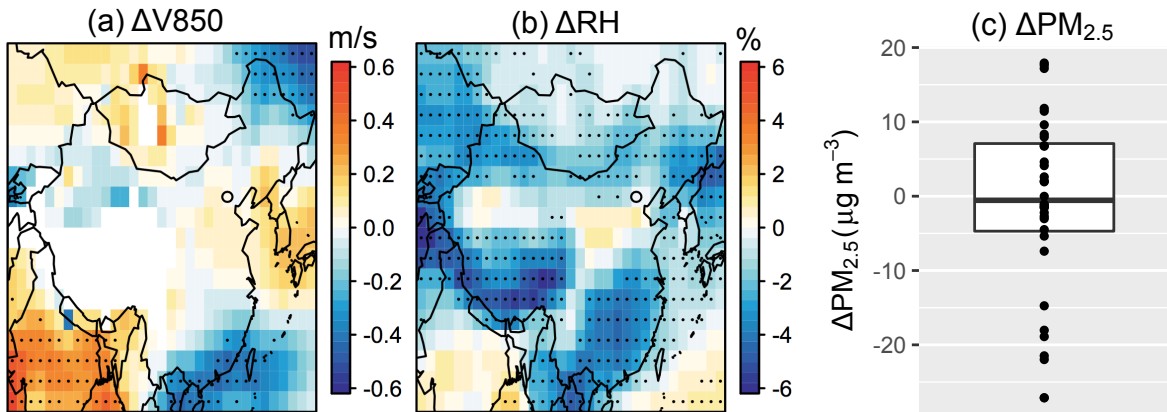

**Figure 5**. Projecting the effect of 21[st]-century climate change (2080-2099 vs. 2000-2019) on Beijing winter haze. Results are for mean wintertime changes ($\Delta$) of (a) meridional wind velocity at 850 hPa (V850) and (b) surface relative humidity (RH) in the RCP8.5 climate change scenario from an ensemble of 32 climate models on a 2°×2.5° grid (Table S2). Gridboxes in which more than 70% of the models show a consistent sign of change are stippled. The location of Beijing is indicated with a circle. (c) Predicted changes of mean wintertime $PM_{2.5}$ in Beijing from individual CMIP5 models (symbols), using the first principal component of V850 and RH (PC1) as a proxy and assuming present-day emissions. The top and bottom of the box are the 25[th] and 75[th] percentile, and the centerline is the 50[th] percentile.

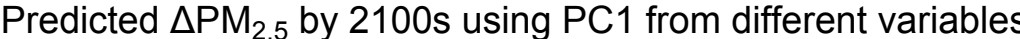

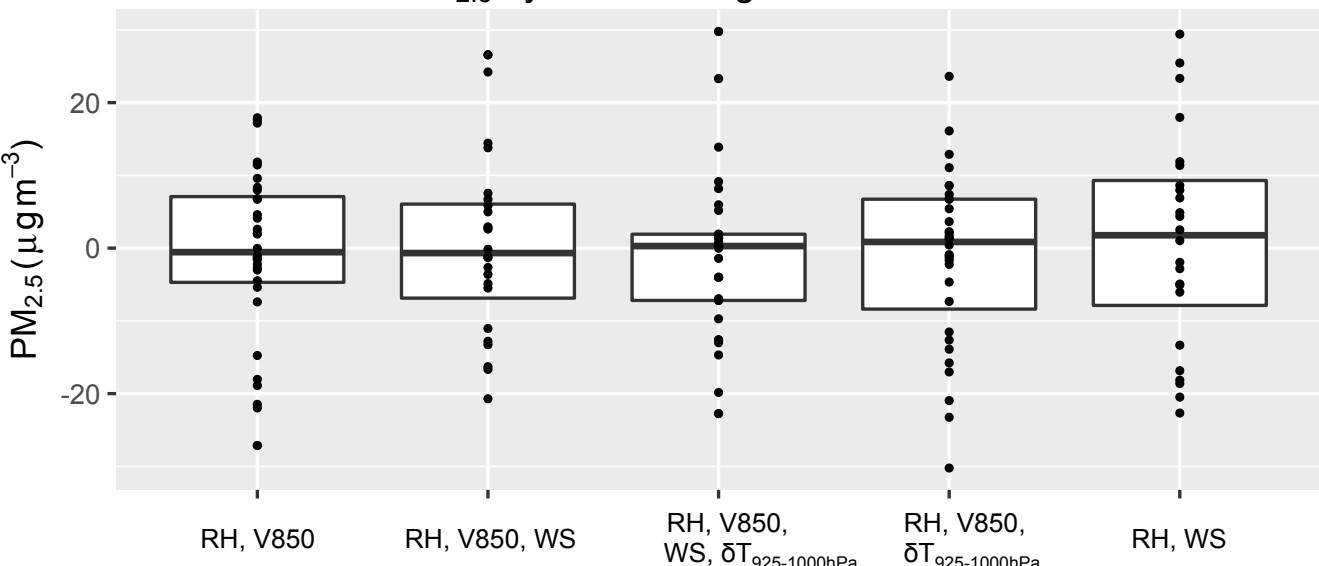

**Figure 6**. Same as Fig. 5c, but using PC1s constructed from five different combinations of meteorological variables as listed in Table S1. We do not include the two PC1s that use $\delta U500$ (experiment #3 and #6 in Table S1). See text for more details.

