# Peer review of "Insignificant effect of climate change on winter haze pollution in Beijing"

_Atmospheric Chemistry and Physics, 2018_

## Referee Comment (RC1) · Anonymous Referee #1 · 18 Oct 2018

This study provides new insights into the impacts of climate change on winter haze in Beijing. I have no doubt that this is an excellent study with significant contribution to the field. The authors show multiple lines of evidence and explain in detail why their results differ from previous studies. I recommend publication in ACP. Below are a few suggestions for consideration by the authors at their discretion:

1. Overall, the manuscript is well structured and the results are clearly presented, but I think the description of the methodology is so condensed that lots of details are not included, which could compromise the reproducibility of this work. For example, how did you construct the principle component for V805 and RH together? Also, it's not clear how you calculated the projected changes in PM2.5. What is exactly the 'PM2.5 vs PC1' relationship in Line 14 of Page 6? Is it calculated based on normalized value

(e.g. Figure 1d) or original value (e.g. Figure 1c)?

2. The authors conclude that the effect of climate change on winter haze in Beijing is likely small based on monthly average PM2.5 data, which reflect the mean state of PM2.5, but it's possible that climate change will have larger impacts on high PM2.5 events as shown in Cai et al., 2017. It's also possible that the distribution of PM2.5 will shift under climate change. I'd suggest the authors comment on how climate change would affect the high PM2.5 and the distribution of PM2.5, despite its insignificant impacts on monthly mean.

3. Figure 2e: The correlation coefficient is calculated based on 7 data points. The positive correlation may be just driven by the high value in 2013. I'm not convinced whether the conclusion is robust with so few data points.

4. Page 2 Line 25: Is observation from a single site regionally representative? Are there any other ground-based observations that can be used? How do they differ?

5. The authors have very detailed supplementary materials, and a lot of their discussions are based on the supplement. I actually spent more time reading the supplementary figures than the main figures, and I think many supplementary figures are worth including in the main text. The balance between supplement and main article is well suited for a letter style article. But for a research article at ACP, I'd suggest the authors consider moving some of important figures from supplement to the main article, so that readers don't have to keep referring to the supplement to follow the discussion.

Minor comments:

Page 4 Line 2: Period missing.

Page 4 Line 27: 'also is also' -> 'is also'.

---

## Referee Comment (RC2) · Anonymous Referee #2 · 30 Oct 2018

This paper studied the reason for the variance in PM2.5 of Beijing during 2010-2017 winters. It shows that V850 and RH can serve as a proxy for Beijing haze and are used to predict the effect of climate change on Beijing haze. The authors make the conclusion different from previous studies. They claim no evidence for a significant effect of climate change on Beijing haze, which sounds really new to me. I recommend publishing the paper after minor revision.

General comments:

1. The authors show that the correlation between PM2.5 and PC1 is larger than V850 or RH alone. Are V850 and RH somehow related in the domain? If so, will it contribute to the larger correlation?

[Figure]

2. The authors put several figures in the supplement, which actually could be very interesting for the readers. I suggest the authors to reconsider the arrangement of figures by moving some important figures to the main text. For example, the figure illustrates the ground measurements.

Specific comments:

1. Page 1, line 13, "than either V850 or RH" alone?

2. Page 3, line 11. I suggest the authors to put the latest reference for NCEP reanalysis here.

3. Please spell it out for "RCP8.5" when the first time used it.

4. Page 6, line 20. The authors pointed out that $\delta$U500 shows no significant trend. However, in the next paragraph, the authors reported a trend in $\delta$U500. It makes me confused here. Please clarify the difference for those two statements.

---

## Author Comment (AC2) · 15 Nov 2018

The comment was uploaded in the form of a supplement:
https://www.atmos-chem-phys-discuss.net/acp-2018-932/acp-2018-932-AC2-supplement.pdf

---

## Author Response (AR1)

**Response to referee comments on "Effect of climate change on winter haze**
**pollution in Beijing: uncertain and likely small"**
We thank the referees for their careful reading of the manuscript and the valuable comments.
This document is organized as follows: the Referee's comments are in *italic*, our responses are in
plain text, and all the revisions in the manuscript are shown in blue. **Boldface blue** text denotes
text written in direct response to the Referee's comments. The line numbers in this document
refer to the updated manuscript.

Reviewer #1 Evaluations:
*This study provides new insights into the impacts of climate change on winter haze in Beijing. I*
*have no doubt that this is an excellent study with significant contribution to the field. The authors*
*show multiple lines of evidence and explain in detail why their results differ from previous*
*studies. I recommend publication in ACP. Below are a few suggestions for consideration by the*
*authors at their discretion:*
Response. Thanks for raising these good points. This feedback has significantly improved the
manuscript. Besides responding to two reviewer's comments, we have also made these changes.
New Title. Insignificant effect of climate change on winter haze pollution in Beijing.
P1 L20. We conclude that climate change is unlikely to significantly offset current efforts to
decrease Beijing haze through emission controls.
*1. Overall, the manuscript is well structured and the results are clearly presented, but I think the*
*description of the methodology is so condensed that lots of details are not included, which could*
*compromise the reproducibility of this work. For example, how did you construct the principle*
*component for V805 and RH together? Also, it's not clear how you calculated the projected*
*changes in PM2.5. What is exactly the 'PM2.5 vs PC1' relationship in Line 14 of Page 6? Is it*
*calculated based on normalized value (e.g. Figure 1d) or original value (e.g. Figure 1c)?*
Response. Thanks. We have included more description of the methodology as suggested by both
reviewers.
P3 L27. PC1 is the sum of V850 and RH after normalization of units:

$$PC1 = \frac{1}{\sqrt{2}}(\frac{V850 - V850_{mean}}{V850_{std}} + \frac{RH - RH_{mean}}{RH_{std}}) \tag{1}$$

where $V850_{mean}$ and $RH_{mean}$ refers to the temporal mean, and $V850_{std}$ and $RH_{std}$ refers to the
standard deviation.
P6 L22-23. Using projected changes in PC1 from the individual CMIP5 models, together with
the $PM_{2.5}$ vs. PC1 relationship of **Fig. 1c**,
*2. The authors conclude that the effect of climate change on winter haze in Beijing is likely small*
*based on monthly average PM2.5 data, which reflect the mean state of PM2.5, but it's possible*
*that climate change will have larger impacts on high PM2.5 events as shown in Cai et al., 2017.*
*It's also possible that the distribution of PM2.5 will shift under climate change. I'd suggest the*
*authors comment on how climate change would affect the high PM2.5 and the distribution of*
*PM2.5, despite its insignificant impacts on monthly mean.*
Response. Thanks for making this good point. We have a companion paper (Pendergrass et al.,
2018, under review in *GRL*) investigating the changes of haze days by 2100s using the extreme
value theory. We have added some discussion in the text.

P7 L3-9. Our results apply to monthly mean $PM_{2.5}$ concentrations, but Pendergrass et al. (2018) used the same meteorological variables with an extreme value point process model to project the effect of climate change on the occurrence of severe wintertime $PM_{2.5}$ pollution events (24-h average > 300 μg m$^{-3}$) in Beijing. They similarly find low V850 and high RH to be the two best predictor variables for these extreme haze events. They show no significant effect of climate change in the frequency of haze events for the RCP8.5 scenario, using the same ensemble of CMIP5 climate models as here, and a decrease for the alternative RCP4.5 scenario due to decreasing RH under low wind speed conditions. They conclude that climate change would most likely decrease the frequency of severe haze events in Beijing.

*3. Figure 2e: The correlation coefficient is calculated based on 7 data points. The positive correlation may be just driven by the high value in 2013. I'm not convinced whether the conclusion is robust with so few data points.*

Response. Thanks. Now we make it clear that we only use a limited history of observations here.

P5 L17-19. Using a 7-year moving average to detrend the timeseries yields a correlation coefficient of 0.69 between reconstructed and predicted PC1 during 1973-2017, and a correlation coefficient of 0.76 between observed $PM_{2.5}$ and predicted PC1 **based on a limited history during** 2010-2017 (Fig. 2e).

*4. Page 2 Line 25: Is observation from a single site regionally representative? Are there any other ground-based observations that can be used? How do they differ?*

Response. Yes, this site is regionally representative. Li et al. (2018) has shown that the correlation of PM2.5 in the embassy site and the average of other 12 sites in Beijing is 0.99. Now we say this in text.

P2 L28-29. Li et al. (2018) have shown that this dataset can be representative of mean $PM_{2.5}$ concentrations in Beijing by comparing with observations from other 12 sites.

*5. The authors have very detailed supplementary materials, and a lot of their discussions are based on the supplement. I actually spent more time reading the supplementary figures than the main figures, and I think many supplementary figures are worth including in the main text. The balance between supplement and main article is well suited for a letter style article. But for a research article at ACP, I'd suggest the authors consider moving some of important figures from supplement to the main article, so that readers don't have to keep referring to the supplement to follow the discussion.*

Response. Thanks. Now we have moved two supplementary figures back to the main text.

[Figure]

Figure 2. Representativeness of relative humidity (RH) observed at Beijing airport and used in this work for correlation to $PM_{2.5}$. (a) Location of Beijing airport and the 12 surrounding sites of the NOAA Global Summary Of the Day (GSOD) network in the North China Plain (NCP). (b) Timeseries of DJF 1973-2017 mean relative humidity (RH) at the Beijing airport site and for the ensemble of the 13 NCP sites. Shaded area denotes one standard deviation. The correlation coefficient (R) of RH at Beijing airport with the North China Plain (NCP) average is 0.92.

[Figure]

Figure 6. Same as Fig. 5c, but using PC1s constructed from five different combinations of meteorological variables as listed in Table S1. We do not include the two PC1s that use δU500 (experiment #3 and #6 in Table S1). See text for more details.

*Minor comments:*
*Page 4 Line 2: Period missing.*
Response. Fixed

*Page 4 Line 27: 'also is also' -> 'is also'.*
Response. Fixed

Reviewer #2 Evaluations:

*This paper studied the reason for the variance in PM2.5 of Beijing during 2010-2017 winters. It*
*shows that V850 and RH can serve as a proxy for Beijing haze and are used to predict the effect*
*of climate change on Beijing haze. The authors make the conclusion different from previous*
*studies. They claim no evidence for a significant effect of climate change on Beijing haze, which*
*sounds really new to me. I recommend publishing the paper after minor revision.*

Response. Thanks for these feedbacks, which has significantly improved the manuscript. Besides
responding to two reviewer's comments, we have also made these changes.
New Title. Insignificant effect of climate change on winter haze pollution in Beijing.
P1 L20. We conclude that climate change is unlikely to significantly offset current efforts to
decrease Beijing haze through emission controls.

*General comments:*
*1. The authors show that the correlation between PM2.5 and PC1 is larger than V850 or RH*
*alone. Are V850 and RH somehow related in the domain? If so, will it contribute to the larger*
*correlation?*
Response. Principal component analysis is usually used to reduce the co-linearity of different
variables. As the reviewer has pointed out, V850 and RH are somehow related, with a correlation
coefficient of 0.60 in our case. The PC1 of V850 and RH thus can be understood as a proxy for
the main weather mode. When compared to either V850 or RH alone, the PC1 includes more
information and is less affected by noises. So it displays higher correlation with $PM_{2.5}$. Now we
say this in text.
P4 L3-4. PC1 includes information from both variables in a way that can better characterize the
main weather mode related to the haze pollution.
P8 L2-3. V850 and RH are only moderately correlated (R = 0.60). Thus PC1 is the better proxy
for the meteorological conditions leading to Beijing haze.
2. The authors put several figures in the supplement, which actually could be very interesting for
the readers. I suggest the authors to reconsider the arrangement of figures by moving some
important figures to the main text. For example, the figure illustrates the ground measurements.
Response. Thanks. Now we have moved two supplementary figures back to the main text.

[Figure]

**Figure 2**. Representativeness of relative humidity (RH) observed at Beijing airport and used in
this work for correlation to PM$_{2.5}$. (a) Location of Beijing airport and the 12 surrounding sites of
the NOAA Global Summary Of the Day (GSOD) network in the North China Plain (NCP). (b)
Timeseries of DJF 1973-2017 mean relative humidity (RH) at the Beijing airport site and for the
ensemble of the 13 NCP sites. Shaded area denotes one standard deviation. The correlation
coefficient (R) of RH at Beijing airport with the North China Plain (NCP) average is 0.92.

[Figure]

**Figure 6**. Same as Fig. 5c, but using PC1s constructed from five different combinations of
meteorological variables as listed in Table S1. We do not include the two PC1s that use $\delta U500$
(experiment #3 and #6 in Table S1). See text for more details.
*Specific comments:*
*1. Page 1, line 13, "than either V850 or RH" alone?*
Response. Now we say "then either V850 or RH alone".

*2. Page 3, line 11. I suggest the authors to put the latest reference for NCEP reanalysis here.*

Response. Thanks. We have checked the NCEP website, and the only suggested reference is Kalnay et al. (1996). So we keep the original reference in text.

*3. Please spell it out for "RCP8.5" when the first time used it.*

Response. Fixed.

*4. Page 6, line 20. The authors pointed out that $\delta U500$ shows no significant trend. However, in the next paragraph, the authors reported a trend in $\delta U500$. It makes me confused here. Please clarify the difference for those two statements.*

Response. Thanks for pointing this out. It is a typo here. Now we say

P6 L27-29. We pointed out previously how alternative principal component predictors of Beijing haze could be derived using additional meteorological variables (i.e.,  $\delta T_{925-1000}$, surface wind speed).